# Transiting Circumbinary Planets in the Era of Space-Based Photometric Surveys

Veselin B. Kostov [1,2] 

1   NASA Goddard Space Flight Center, 8800 Greenbelt Road, Greenbelt, MD 20771, USA; veselin.b.kostov@nasa.gov
2   SETI Institute, 189 Bernardo Ave, Suite 200, Mountain View, CA 94043, USA

**Abstract:** Planets orbiting binary stars—circumbinary planets—play a paramount role in our understanding of planetary and stellar formation and evolution, dynamical interactions in many-body systems, and the potential for habitable environments beyond the Solar System. Each new discovery holds immense value and inherent fascination both for the astronomical community and for the general public. This is perhaps best demonstrated by the 1500+ citations of the discovery papers for the 14 known transiting circumbinary planets and the dozens of related press-releases in major news outlets. This article reviews the observational and theoretical aspects related to the detection and confirmation of transiting circumbinary planets around main-sequence binaries from space-based surveys, discusses the associated challenges, and highlights some of the recent results.

**Keywords:** circumbinary planets; eclipsing binary stars; transits; Kepler; TESS

## 1. Introduction

One of the major obstacles for detecting exoplanet transits is the a priori unknown orbital inclination of the system. A star may host multiple planets, but if they never cross the line of sight with the observer, the latter can never see their transits. Thus, astronomers have to monitor a large number of stars and statistically beat the geometric odds the universe has stacked against them (e.g., [1,2]). And spectacularly beat them we have—at the time of writing, there are thousands of confirmed and candidate transiting exoplanets discovered with dozens of different telescopes and observatories. See https://exoplanetarchive.ipac.caltech.edu for an up-to-date number of confirmed and candidates planets. Note that these include planets according to the criteria listed at https://exoplanetarchive.ipac.caltech.edu/docs/exoplanet_criteria.html. In fact, despite the geometric odds, transit detection has so far been the most productive method to discover exoplanets.

Long before the discovery of the first transiting exoplanet (HD 209458 b [3]), it was realized that the sample of target stars can be pre-selected for a favorable geometric configuration by observing eclipsing binaries (EB) instead of single stars (e.g., [2,4–9]). As EBs are already nearly aligned with our line-of-sight, so should the orbits of their circumbinary planets (CBPs) be—close to edge-on from the observer's perspective—which would dramatically increase the probability of detecting transits. In addition, consecutive transits of a CBP typically vary in depth, duration, and shape, and can noticeably deviate from a linear ephemeris. This is a unique observational signature without false positives that immediately provides a strong planet candidate. It is worth noting that triply-eclipsing triple stars that produce tertiary eclipses and/or occultations (e.g., [10]) exhibit a similar observational signature. However, radial velocity measurements and/or detailed photometric-dynamical analysis for these systems can rule out planetary interpretation for tertiary events detected in the photometric data.

Extensive theoretical studies have shown that the formation of CBPs should be a robust and common process, producing a variety of planets and orbital configurations

(e.g., [11–41]). Similar to single-star systems, CBPs form further out in the circumbinary protoplanetary disk surrounding the host binary star and then are thought to migrate towards the inner edge of the disk. During their migration, the CBPs have to navigate through multiple orbital resonances with the binary star and, depending on the physical and orbital parameters of the system, may become unbound or even collide with a star (e.g., [29,42]). The planets that survive the complex dynamical gauntlet of close binary stars attain long-term dynamically stable orbits that are larger than a minimum critical distance from the binary $a_{crit}$ (e.g., [17,43–51]). The latter is a function of the binary separation $a_{bin}$ and depends on the orbital eccentricity, inclination, and binary mass ratio. For co-planar orbits $a_{crit} \sim 2.5$–$4.5\ a_{bin}$ (e.g., [45], and references therein), and the corresponding critical minimum period $P_{crit}$ for long-term dynamical stability of a CBP is $\sim 4$–10 times longer than the binary period $P_{bin}$. Last but not least, the occurrence rate of giant planets is expected to be similar between single stars and close binary stars, i.e., on the order of 10% (e.g., [52–56]).

It is important to note that CBPs represent a subclass of planets residing in binary star systems and are generally labeled as "P-type" planets (e.g., [57]). Planets orbiting around either star in a wide binary system are labeled as "S-type", and planets orbiting around the L4 or L5 Lagrange points of a binary system are labeled as "L-type". To be long-term dynamically stable, S-type planets must have semi-major axes smaller than about 1/3 the binary separation (e.g., [45], and references therein).

The existence of protoplanetary circumbinary disks was firmly established at the turn of the century (e.g., [39,58], and references therein), with dozens of circumbinary disks resolved or inferred at the time of writing. These cover a wide range of physical and orbital parameters, including co-planar or inclined (even polar), warped, torn, and very young disks (e.g., [59–69], and references therein). Combined with a high expected frequency of circumbinary disks around compact young systems (almost 90% for binaries with semi-major axes smaller than 1 au [39]), these discoveries demonstrate that the necessary ingredients for planet formation around binary stars are readily available.

Altogether, these considerations make detached EBs with orbital periods from days to a few weeks excellent targets for finding transiting planets. The search has been ongoing for decades, with early efforts focused on detecting small planets around small stars. Refs. [5,7] argued that photometric observations of bright, nearby K- and M-dwarf close binaries with a ground-based network of 1-m class telescopes would be able to detect transiting Earth-sized CBPs. For example, the Transits of Extrasolar Planets (TEP) project [9,70] monitored the CM Draconis system for CBP transits between 1994 and 1999. This is a particularly well-suited EB for a CBP search due to its short orbital period ($\approx 1.27$ days), small stars ($M_A = 0.23\ M_\odot$, $M_B = 0.21\ M_\odot$, $R_A = 0.25\ R_\odot$, $R_B = 0.24\ R_\odot$) and nearly edge-on orbit (i = 89.77 degrees) [71,72]. While the TEP observations were sensitive enough to detect planets as small as 3 $R_\odot$ with 90% probability, no transits were detected [73,74]. This did not discourage CBP hunters. Quite on the contrary, the search continued and, thanks to observations from the Kepler [1] and TESS [75] missions, culminated with the successful discovery and confirmation of transiting CBPs around main-sequence binary stars—the main focus of this review. Several CBP candidates have been detected around evolved EBs as well, based on eclipse timing variations (e.g., [76], and references therein).

TESS has been observing CM Draconis since August 2019. At the time of writing, there are 16 sectors of data available, 15 of which are in short-cadence, covering a baseline of nearly 1200 days. Figure 1 shows the short-cadence TESS data from Sector 56. The prominent primary and secondary eclipses for the two stars are easily identified, as well as the stellar activity in the form of out-of-eclipse lightcurve modulations and flaring events. The figure also highlights the expected transit depths for Jupiter- and Neptune-sized CBPs, indicating that such transits should be easily identifiable even to an untrained eye. A closer inspection of all currently-available TESS data shows no obvious transits. With that said, TESS will continue to observe CM Draconis until September 2024 so the system may still have a surprise in store for us.

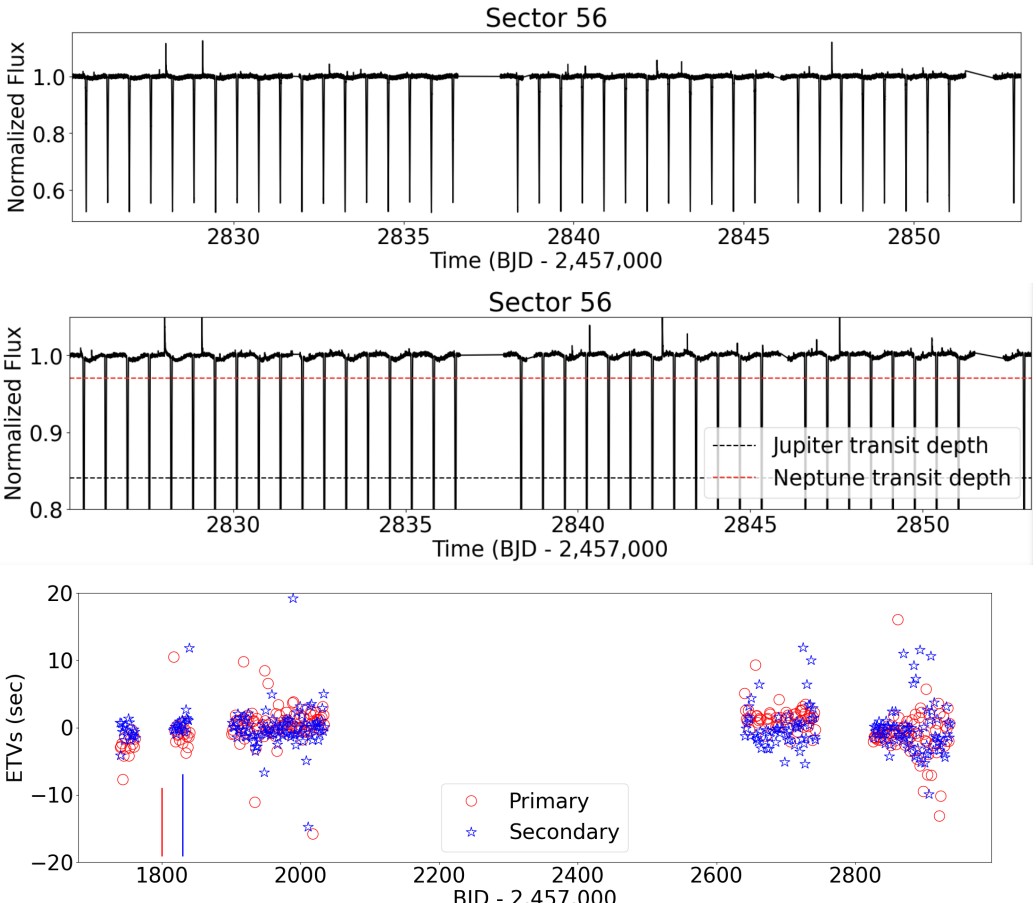

**Figure 1. Upper panel**: Short-cadence TESS data for CM Dra. **Middle panel**: same as the upper panel but zoomed-in to highlight the expected transit depth for a Neptune-sized CBP (red dashed line) and a Jupiter-sized CBP (black dashed line). **Lower panel**: measured eclipse timing variations for the primary (red) and secondary (blue) eclipses. Both eclipses follow linear ephemeris. The vertical red and blue lines represent the typical per-point uncertainties.

## 2. Transiting Circumbinary Planets from Kepler and TESS

Planets orbiting main sequence binary stars have long been an integral part of science fiction, but a scientific proof of their existence has been elusive until the discovery of Kepler-16 b [77]. What made this planet special and its discovery unambiguous was that it produced transits—both across the primary and the secondary star—during each conjunction observed by Kepler. This is in contrast to other candidates where eclipse timing variations were interpreted as light-travel time delays caused by a potential circumbinary planet. For the case of Kepler-16 b, Ref. [77] observed dramatic transit timing variations (on the order of several days), duration variations (several hours), and depth variations between consecutive conjunctions—the "smoking gun" signature of a transiting CBP [23]. The authors reproduced these to a high degree of precision and accuracy with a comprehensive circumbinary model based on numerical integrations (e.g., [78,79]), making the case for the existence of main sequence CBPs indisputable.

The discovery of CBPs was indeed one of the major breakthroughs from the Kepler and TESS missions. Kepler-16 b opened the door for more CBP detections and following right on its heels were two more confirmed transiting planets in orbit around EBs, Kepler-34 b and Kepler-35 b [80]. Within months, Kepler-38 b [81], the first transiting multiplanet circumbinary system Kepler-47, and the first CBP in a quadruple stellar system were announced [82–84]. Interestingly, only the inner and outer planets of Kepler-47 were detected in 2012, and the third planet was added a few years later. Soon after came the discovery of Kepler-413 b, a rather peculiar CBP that "missed" transits more often than it

"hit" due to rapid orbital precession [85]. The transits of Kepler-413 are not only aperiodic and vary in depth and duration, but also ceased altogether for about 800 days shortly after the beginning of the Kepler observations, only to reappear again near the end of the observations. Such an unusual behavior of transiting CBPs was in fact predicted more than 20 years before the discovery of Kepler-413 b [6]. With all of these discoveries, a new class of planets was firmly established.

Altogether, fourteen transiting planets in orbit around twelve EBs have been discovered and confirmed since 2011, twelve from Kepler and two from TESS (e.g., [86], and references therein). Two non-transiting CBPs around main-sequence binaries have been discovered as well: one from Kepler data based on eclipse timing variations [87–89] and another from radial velocity observations of a TESS CBP system [90]. A non-transiting circumbinary brown dwarf was also discovered around HD 202206 from radial velocity observations (e.g., [91]). This rapid chronological progress in the field of CBPs also represents a dramatic progression in terms of the complexity and diversity of the planetary systems found. These provide observational tests for theoretical predictions on the formation and evolution of CBPs, shed light on the rich dynamical environments of close binary stars, and even provide new insight into how these binaries form. Several intriguing features of transiting CBPs have already emerged, as discussed below.

- **Orbital Configuration and Observability** The current family portrait of CBPs discovered from Kepler and TESS is shown in Figure 2 in terms of orbital period, distance, planet size, and distance from the critical limit for long-term dynamical stability. As seen from the figure, all except Kepler-1647 b [92]) and KIC 7821010 are very close to their host binary stars—within a factor of two of the critical separation for stability $a_{crit}$ (e.g., [45,49,57])—with Kepler-16 b orbiting just ∼10% away from $a_{crit}$.

  The stability limit has a direct consequence for the detection of transiting CBPs and is responsible for one of the major observational challenges related to these planets. As highlighted in the left panel of the figure, only one of the planets has an orbital period shorter than 50 days (Kepler-47 b at 49.5 days), and another has an orbital period of about 66.3 days (Kepler-413 b). The rest have orbital periods of hundreds of days, reaching up to nearly four years for the case of Kepler-1647 b. Even if we ignore the complications presented by the above-mentioned aperiodicity and precession-induced missed transits, finding such orbital periods from the ground is exceedingly difficult (even practically impossible). Thus, it is not surprising that the first detection and confirmation of a transiting CBP had to wait for the nearly continuous, years-long observational coverage provided by the Kepler mission.

  Another major observational challenge is that transiting CBPs do not necessarily transit during every conjunction. Unlike single-star planets that either transit over the course of the observations or do not, CBPs may transit or may not. Due to the orbital precession on relatively short timescales a CBP can produce a complex pattern of potentially sporadic transits. This is highlighted in Figure 3 for the case of Kepler-413 b where the precession period is ∼11 years [85]. Depending on the orbital configuration and dynamical complexity of the system, a CBP can produce transits during non-consecutive conjunctions, start/stop transiting during the observations or even start, then stop, and then start again [52,53,85]. Importantly, this is not a rare occurrence as several CBP systems exhibit this unusual observational signature during observations from both Kepler and TESS, and the known CBPs produce transit during only about ∼5–10% of all conjunctions (e.g., [23]).

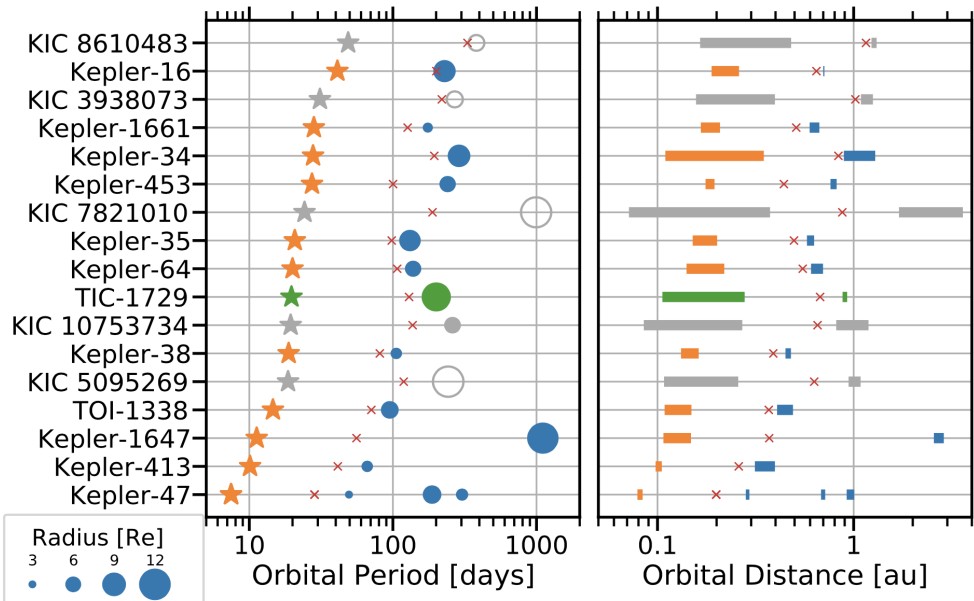

**Figure 2.** Taken from [86]. Planet radius, orbital period and distance from the host binary star for the known transiting (filled orange, green, and gray symbols for the host binaries, and circles for their planets) and non-transiting (open symbols, detected through eclipse-timing variations) CBPs discovered from Kepler and TESS data. The horizontal bars in the right panel show, on a logarithmic scale, the eccentricity-modified orbital separation of the planets from the binary (bars at large distance) and of the binary itself (bars at smaller distance). The red crosses on the horizontal lines represent the respective stability limits.

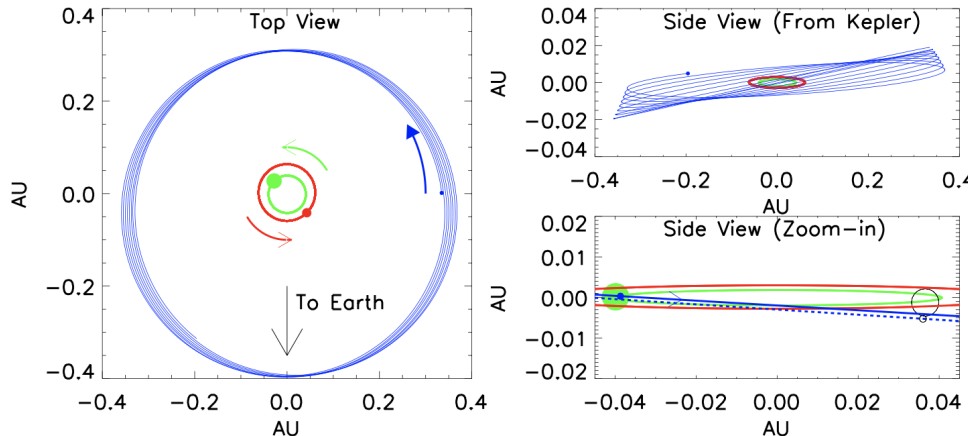

**Figure 3.** Taken from [85]. To-scale orbital configuration of the Kepler-413 system highlighting the rapid orbital precession of the CBP (precession period of ∼11 years). The solid overlapping symbols in the lower right panel represent the configuration of the system during the last transit observed from Kepler, and the open circles represent a missed transit one CBP conjunction later.

It is also important to note that the absence of transits in the lightcurve of an EB does not necessarily correspond to the absence of potential planets as these could simply transit during data gaps (similar to single-star planets) or during stellar eclipses. If a transit is completely blended with an eclipse, i.e., a syzygy, recovering the former would require nearly perfect removal of the former. In fact, to account for this eventuality, every eclipse needs to be carefully removed from the lightcurve as the transit time is a priori unknown. This is a complex and time-consuming effort that requires thorough analysis of the intrinsic variability of both stars and could produce a large number of false positives. If, however, the blend is partial and the signal to noise is

sufficient, it might be possible to identify the transit without extensive manipulation of the lightcurve. This is highlighted in Figure 4 for the case of Kepler-1647 b [92].

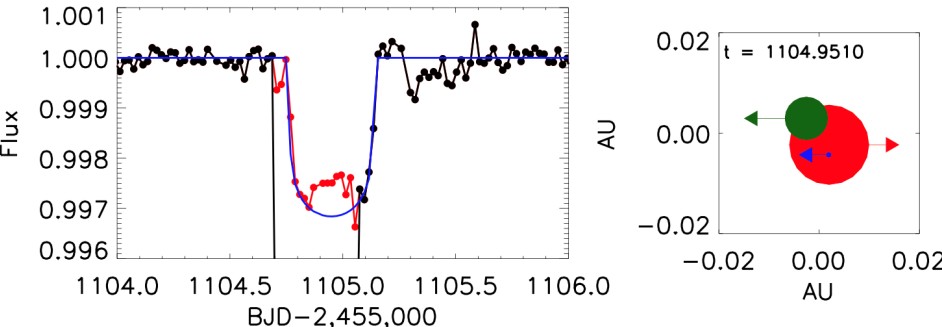

**Figure 4.** Taken from [92]. **Left panel**: A section of the Kepler lightcurve (small dots) during a CBP syzygy with a stellar eclipse. The eclipse is the deep feature that falls below the x-axis as it is too deep to fit on the scale of the panel. The red symbols represent the in-transit data and the blue curve represents the transit model. **Right panel**: Corresponding configuration of the system as seen from Kepler.

- **CBPs vs. single-star planets:** An interesting question to consider is whether the single-star and close-binary-star planets exhibit common characteristics. While addressing this question likely requires a considerably larger sample than 14 planets, a qualitative comparison between the two populations could uncover valuable insights that merit further exploration. Figure 5 illustrates such comparison in terms of the radius, period, and insolation for the confirmed transiting planets around EBs, around single stars (including the Solar System), and around one star in a multistar system (S-type configuration, [44]). Given the associated observational biases, sample sizes and completeness, the CBPs seem to be larger, receive less insolation, and reside on wider orbits.

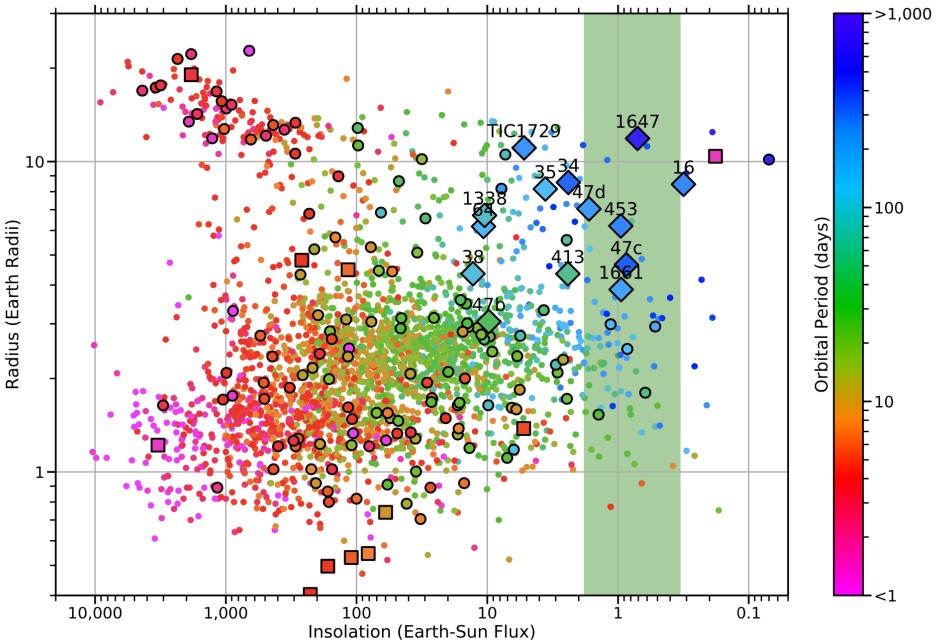

**Figure 5.** Taken from Kostov et al. (2021). Planet radius, orbital period, and insolation for transiting CBPs (diamond symbols), confirmed single-star planets (small dots), planets in S-type orbits in wide binary systems (circles) and wide higher-order systems (squares). The vertical green band represents the regime of the habitable zone.

Whether these are representative characteristics of the general CBP population is still unclear, yet there are certain aspects worth considering. While the 12 known CBP systems seem to occupy a distinct parameter space in Figure 5, there are hints of an underlying diversity with regard to their physical and orbital properties. In particular, five systems already stand out from the rest: (i) Kepler-16 due to its proximity to the stability limit ($a_{cbp} \sim 1.1 a_{crit}$); (ii) Kepler-34 due to its high eccentricity ($e_{cbp} = 0.52$); (iii) Kepler-47 and TOI-1338 due to their multiplicity (3- and 2-planet systems, respectively); (iv) Kepler-413 due to its rapid orbital precession (precession period of $\sim$12 years); (v) and Kepler-1647 due to its long orbital period ($P_{cbp} = 1107$ days).

Four of the transiting CBPs are in the conservative habitable zone [93]. In terms of habitability, CBP are particularly interesting targets as the insolation they receive is far from constant. Instead, it varies on multiple timescales depending on the binary period, the CBP period, and the CBP precession period (e.g., [94–103], and references therein). The variations can be substantial, even to the point of a CBP moving into and out of the habitable zone during a single planetary orbit. In addition, one has to take into account the insolation contribution from two stars of different spectral types, the response of the atmosphere to the (potentially) rapid variations, the evolution of the stars, and various other factors. Furthermore, binary systems with orbital periods longer than 10 days may in fact be better in terms of habitability prospects compared to single stars as tidal interactions between the stars can decrease chromospheric activity (potentially drastically) that might otherwise be harmful for the emergence and evolution of life [104,105].

Additionally, while today we know of thousands of planets outside the solar system, the situation was quite different twenty five years ago. At the time, the sample was much smaller—not unlike the current state of CBPs. This is highlighted in Figure 6 showing the orbital period as a function of orbital eccentricity of the transiting CBPs in 2023 (star symbols) and of all exoplanets in 1998 (circles). One can compare the latter to the number of exoplanets in 2023 and easily imagine the enormous missed opportunity if we stopped searching for these in 1998. Thus, as exciting as the currently-known transiting CBPs are, they are just the tip of the iceberg.

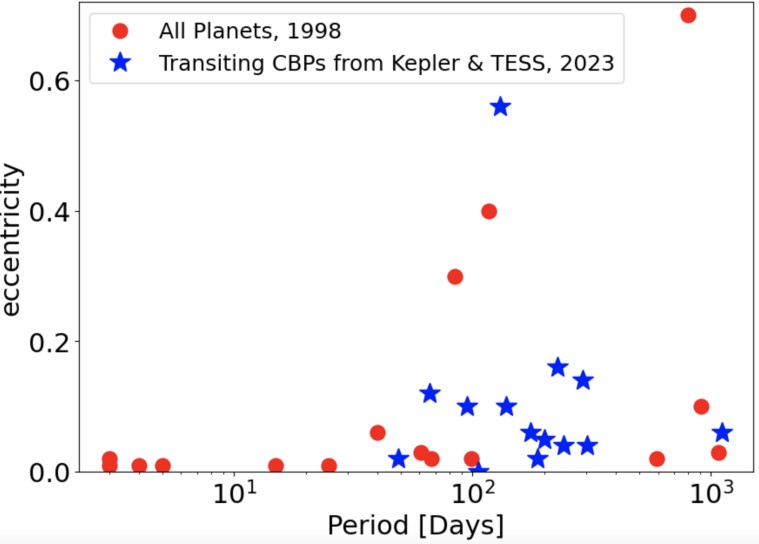

**Figure 6.** Comparison between the orbital period and eccentricity of all exoplanets known in 1998 and of all transiting CBPs known in 2023.

Finally, it is worth noting that it took about two and a half years between the launch of Kepler and the discovery of the first transiting CBP from the mission (Kepler-16 b), but only about half as much between the launch of TESS and its first transiting CBP (TOI-1338 b). This demonstrates that CBP hunters are becoming better at catching

their prey, and suggests that new and exciting discoveries are awaiting. Also, Kepler-1647 b and Kepler-1661 b were announced years after the Kepler spacecraft stopped observing, so there may still be CBPs waiting to be found in the archival data.

- **Host systems** Within the general EB population, the binary hosts of the transiting CBPs are typical in terms of their stellar masses, mass ratios, orbital periods, eccentricities, effective temperatures, and ages (e.g., [23,56,106]). Interestingly, the estimated metallicity for most of the host systems is Solar or sub-Solar, with only Kepler-64 and TIC-172900988 having higher values. This seems to be in contrast to single-star systems where the occurrence frequency of planets larger than about $4R_\oplus$—the radius regime of the transiting CBPs—increases with metallicity (e.g., [107]). However, given the associated uncertainties with the metallicity estimates of the CBP systems, it is unclear if this trend is statistically significant. The only reason the CBP hosts seem to stand out from the rest of their siblings is the presence of the planets.

  To date, no transiting CBPs have been found orbiting around EBs with an orbital period $P_{EB} < 7.5$ days even though most of the Kepler and TESS EBs have much shorter orbital period (e.g., [23,106], and references therein). Specifically, two out of three Kepler EBs have $P_{EB} < 7.5$ days, with half having an orbital period shorter than 2 days [108,109]. Assuming CBPs around EBs with $P_{EB} < 7.5$ are as close to the stability limit as those around EBs with $P_{EB} > 7.5$ and also nearly co-planar, their comparably shorter orbits should make them easier to find as they would exhibit more transits per unit time. In other words, while there is an observational bias against longer-period CBP, there is an observational preferences for shorter-period CBPs. With that said, the lightcurves of short-period EBs are generally more difficult to analyze due to stronger intrinsic variability, more stellar eclipses per unit time, which leaves less out-of-eclipse data to search through, etc. It is worth noting that [110] devised a method to find CBPs around short-period EBs by detecting the reflected light of the binary on the planet.

  The absence of CBPs around Kepler EBs with $P_{EB} < 7.5$ days was also corroborated by a non-detection of CBPs in the CoRoT EB sample. The latter had a similar size to Kepler's and even though the coverage of CoRoT was shorter, the quality of the data would have been sensitive enough for the detection of such CBPs [111]. This apparent paucity of transiting CBPs in short-period EBs has been attributed to the evolutionary history of close binary stars. Specifically, such short-period EBs could not have formed at their current orbital configurations, but likely arrived at them through Kozai–Lidov interactions with a distant tertiary stellar companion [112–115]. In turn, this would have inhibited the formation of CBPs or destabilized them after they formed (e.g., [42,116–118]).

- **High-precision measurements:** While stellar eclipses occur at (nearly) fixed orbital phases, the transits of a CBP can occur at practically any phase of the host binary and thus probe the system at many different orbital arrangements. These "snapshots" provide additional constraints on the sizes, orbital positions, and relative velocities of the stars and the planet that, in turn, enable exquisitely precise measurements of their properties. Furthermore, the transit chord a CBP traverses across the host star(s) is typically different for different conjunctions, which can further improve the treatment of the stellar limb darkening.

  The precision and accuracy aspects hold particular significance for CBP systems that contain low-mass stars where the measurements can help address the long-standing discrepancy between their observed and predicted sizes (e.g., [119]). Some of the most precisely measure masses and radii of low mass stars have indeed been obtained from CBP systems. As an example, the M-dwarf secondary of Kepler-16 has $M_B = 0.2026 \pm 0.0007 M_\odot$ and $R_B = 0.2262 \pm 0.0008 R_\odot$, and the CBP has a mass and radius uncertainty of $\approx 0.016 M_{Jupiter} \approx 0.0025 R_{Jupiter}$, respectively (Doyle et al., 2011 [77]). The measurements for the Kepler-34 and Kepler-35 systems are equally impressive: $M_A = 1.0479 \pm 0.0033 M_\odot$ and $R_A = 1.1618 \pm 0.0030 R_\odot$, $M_B = 1.0208 \pm 0.0022 M_\odot$

and $R_B = 1.0927 \pm 0.0030R_\odot$ for the former, and $M_A = 0.8877 \pm 0.0052M_\odot$ and $R_A = 1.0284 \pm 0.0020R_\odot$, $M_B = 0.8094 \pm 0.0043M_\odot$ and $R_B = 0.7861 \pm 0.0021R_\odot$ for the latter [80]. Such remarkable precision and accuracy are achieved by reproducing all available photometric and spectroscopic data, including radial velocity measurements and follow-up observations, with a comprehensive photodynamical model (e.g., [78–80]).

- **Detection Methods** Traditional methods for detecting transiting planets around single stars by phase-folding the data on a fixed period are inadequate for finding the transits of a CBP. Folding the lightcurve of a CBP on the best-fit period will dilute the transits instead of building up the signal-to-noise. This is highlighted in Figure 7 for the case of Kepler-16 b, where consecutive transits are early or late by days compared to a linear ephemeris.

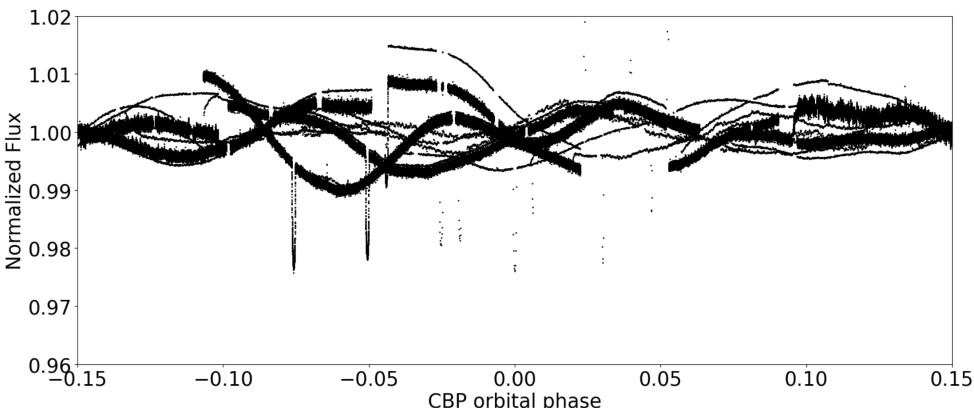

**Figure 7.** Kepler lighturve of Kepler-16 phase folded on the best-fit period of the CBP ($P_{CBP}$ = 228.776 days). Folding the data on $P_{CBP}$ smears the transits in phase space.

All fourteen transiting CBPs from Kepler and TESS have been discovered by visual inspection of EB lightcurves. The lightcurves are typically prepared for inspection by, e.g., detrending low-frequent flux-variations, folding on the period of the binary, removing of the stellar eclipses. While this approach may seem outdated in the age of machine-learning algorithms and is certainly time-consuming, it has been remarkably successful and productive. Several automated detection methods for transiting CBPs have been developed over the years to tackle their complex observational signatures (e.g., [8,52,82,111,120–123]). With the exception of the semi-automated procedure of [82], these have yet to detect CBP transits before the human eye does. Nevertheless, the algorithms recover the known CBPs (e.g., [52,122,123]), so it seems the contest between humans and machines is far from over.

Naturally, visual identification is entirely dependent on a particular CBP transit being of sufficiently high signal-to-noise to stand out to the human eye from the rest of the target-specific features in the lightcurve. There are many obstacles towards the discovery of a CBP, such as prominent out-of-eclipse stellar variability, insufficient data, various systematics (whether astrophysical or instrumental), and false positives. The latter two are particularly irksome, as it can be quite disappointing to find the proverbial needle in the celestial haystack of lightcurves after an extensive search only to realize it is a dud. As an example, Figures 8–10 highlight several such disappointments, all found by visual inspection (priv comm), that masquerade as transiting CBPs. These represent some of the ongoing efforts to find transiting CBPs in TESS data, and demonstrate that relatively shallow, transit-like events can be readily identified in the lightcurves of EBs observed by the mission.

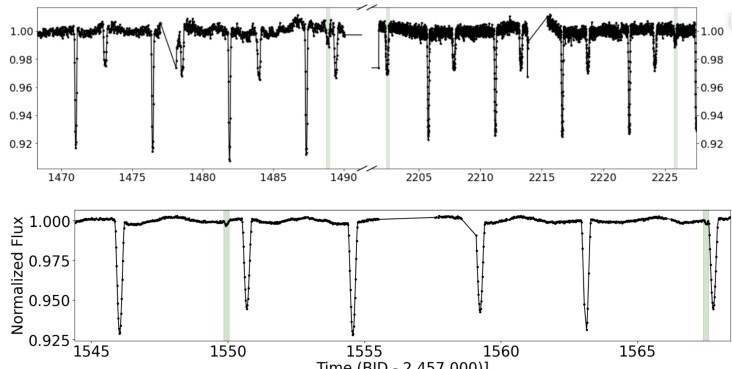

**Figure 8.** Examples of CBP false positives detected by visual inspection of TESS EB lightcurves. **Upper panel**: Taken from [124]. Normalized TESS ELEANOR data [125] of the stellar quadruple candidate TIC 438226195 from Sectors 6 and 33. The vertical green bands highlight extra transit-like events in addition to the clear EB signal. Initially, only Sector 6 data were available, exhibiting a single extra transit-like event in the TESS ELEANOR data [125]. Subsequent data from Sector 33 showed that the event is in fact an eclipse from a second EB, making TIC 438226195 an on-target CBP false positive. **Lower panel**: TESS QLP data [126] of the CBP false positive TIC 92469903 from Sector 9 showing two extra transit-like events.

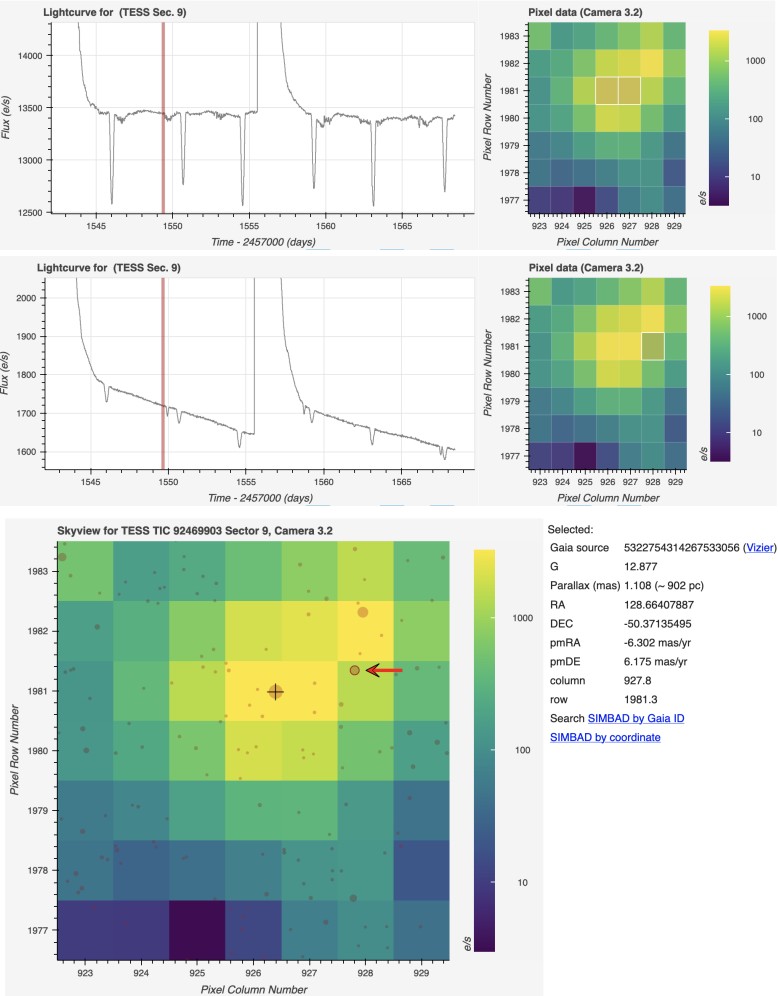

**Figure 9.** CBP false positive TIC 92469903. **First and second panel**: Lightcurve and pixel-level analysis of TESS data showing that the source of both extra events is the nearby EB TIC 92469882. **Last panel**: 7 × 7 pixels Skyview image of the field around TIC 92469903 highlighting the contaminator TIC 92469882.

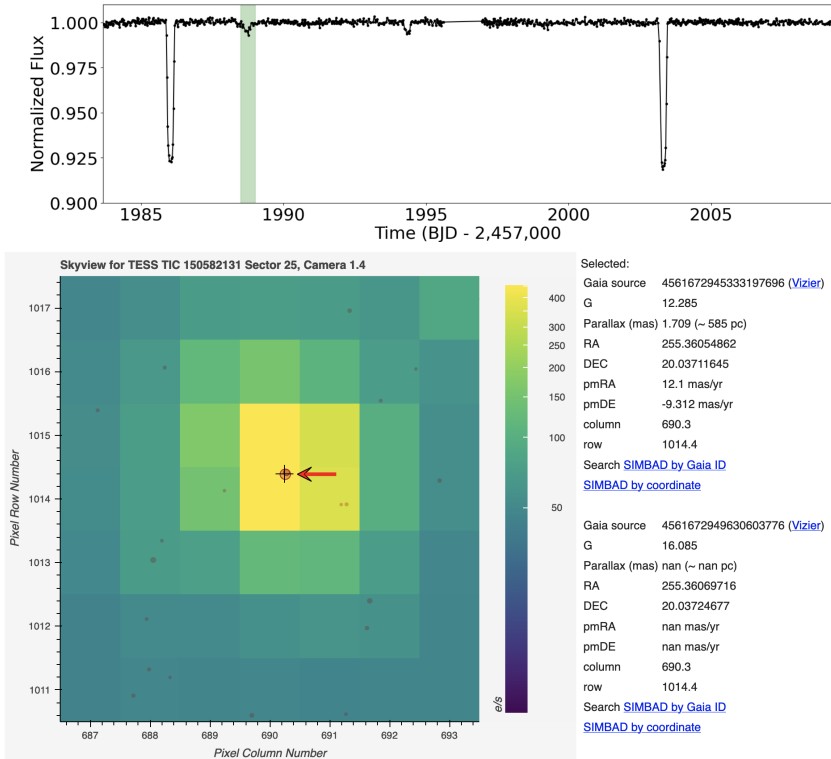

**Figure 10.** Likely CBP false positive TIC 150582131. **Upper panel**: TESS ELEANOR lightcurve of TIC 150582131 from Sector 25 showing one extra event. **Lower panel**: 7 × 7 pixels Skyview image of the field around TIC 150582131 showing a resolved, closely-separated (0.4 arcsec) star (TIC 508200354). The separation between the two stars is too small to determine which of them is the source of the extra event based on the available data, and the magnitude difference is such that either can produce the extra event. This makes TIC 150582131 a likely CBP false positive.

## 3. Looking into the Future

Extrapolating from the remarkable progress in the field of transiting CBPs over the past decade, it is not unreasonable to foresee a bright future ahead of us. We can expect exciting new CBP discoveries thanks to (i) the observatories we already have (such as TESS); (ii) those we will soon be able to use, such as the PLAnetary Transits and Oscillations of stars mission (PLATO, [127], with an expected CBP yield 3–4 larger than Kepler's, see https://indico.ict.inaf.it/event/806/contributions/4313/attachments/2657/5210/20190925_Deeg.pdf) and the Nancy Grace Roman Space Telescope; (iii) the mostly-untapped potential of archival data from, e.g., K2 [128]; and (iv) the ever-improving detection and analysis tools, methods and techniques. For example, the K2 mission, in particular, observed a wider diversity of stellar types compared to the original Kepler mission, with a much greater focus on late-type stars. This provides the opportunity to discover CBPs in a variety of environments, such as at different Galactic latitudes, stellar populations (e.g., open clusters). And while the K2 lightcurves generally have a lower SNR compared to those from Kepler, their photometric quality is still remarkably good. Specifically, the median Combined Differential Photometric Precision of K2 is ∼80 ppm at a magnitude of Kp = 12 mag in a 6-h window, and about 10% of all K2 targets have a SNR of less then 50 ppm. Thus, the K2 archive is more than adequate for finding the transits of Kepler-like CBPs, and provides an excellent opportunity to find CBPs smaller than $4R_\oplus$ around Sun-like close binary stars.

Capitalizing on these discoveries, we will have the opportunity to tackle the still-open questions addressing the (i) formation and evolution pathways of these planets and their host binaries; (ii) formation efficiency and occurrence frequency; (iii) underlying population characteristics; and (iv) viable orbital architectures.

**Funding:** This research was partially funded by NASA grant number 80NSSC22K0747.

**Data Availability Statement:** The data presented in this study are openly available at MAST.

**Conflicts of Interest:** The author declares no conflict of interest.

## Abbreviations

The following abbreviations are used in this manuscript:

EB      Eclipsing Binary
CBP     CircumBinary Planet
TESS    Transiting Exoplanet Survey Satellite

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
