# Peer review of "Transiting Circumbinary Planets in the Era of Space-Based Photometric Surveys"

_universe, doi:10.3390/universe9100455_

Round 1
Reviewer 1 Report
The Kostov paper is a review of observational and theoretical aspects related to the detection and confirmation of circumbinary planets. Under this aspect, it is a quasi encyclopaedia, given its impressive bibliography and deserves publication after the following few changes :
1/ In the footnote 1, first page, it gives a precise number of confirmed planets as of 20 Sep 2023, according to https://exoplanetarchive.ipac.caltech.edu . The problem here is that this archive includes only planets with “mass (or minimum mass) is equal to or less than 30 Jupiter masses”,
with no argument (https://exoplanetarchive.ipac.caltech.edu/docs/exoplanet_criteria.html ),
whereas Hatzes and Rauer develop an argument fin favor of an upper mass around 60 Jupiter masses (https://doi.org/10.1088/2041-8205/810/2/L25 ).
2/ In the footnote 2, first page, it states that « Other than a triply-eclipsing triple star, which can be ruled out from the radial velocity measurements. ».
Is this the candidate planet GW Ori b ?
(https://ui.adsabs.harvard.edu/abs/2021MNRAS.508..392S/abstract )
I have not found in the literature any paper excluding GW Ori b from RV measurements.
For completeness, it would be useful that the author gives the candidate circum-triple planets name and a reference for its rebuttal.
Can the author clarify this point?
3) From a general point of view, since the paper intends to be a general review, I make some suggestions:
i) Circumbinary planets are a subclass, generally named “P-type”, of planets in binary star systems, planets orbiting only one star in a wide binary being of “S-type”. The author could place circumbinary planets in perspective in this scheme, and remind that the planetary stable orbits must have a semi-major axis < a/3 for S-type planets and > 3a for P-type planets, where a is the the separation between the two stars (for equal mass stars)
ii) The author could add the method of selecting stars to search for transits by Hale and Doyle:
https://ui.adsabs.harvard.edu/abs/1994Ap%26SS.212..335H/abstract
iii) In the spirit of the general review aspect of the paper, the author could add a few paragraphs on habitability of circumbinary planets: e.g. papers by Eggl and by Haghighipour
Author Response
Oct 15, 2023
Dear reviewer,
Thank you for the detailed feedback. I have revised the manuscript "Transiting Circumbinary Planets in the Era of Space-Based Photometric Surveys", following closely your suggestions.
Please find below the detailed response to your comments. In the revised manuscript, I have set off updated or new text in bold faced font for your convenience.
Sincerely,
Veselin Kostov
--------------
The Kostov paper is a review of observational and theoretical aspects related to the detection and confirmation of circumbinary planets. Under this aspect, it is a quasi encyclopaedia, given its impressive bibliography and deserves publication after the following few changes :
# # #
1/ In the footnote 1, first page, it gives a precise number of confirmed planets as of 20 Sep 2023, according to https://exoplanetarchive.ipac.caltech.edu . The problem here is that this archive includes only planets with “mass (or minimum mass) is equal to or less than 30 Jupiter masses”,
with no argument (https://exoplanetarchive.ipac.caltech.edu/docs/exoplanet_criteria.html ),
whereas Hatzes and Rauer develop an argument fin favor of an upper mass around 60 Jupiter masses (https://doi.org/10.1088/2041-8205/810/2/L25 ).
--> Response to reviewer: Footnote 1 has been updated accordingly.
# # #
2/ In the footnote 2, first page, it states that « Other than a triply-eclipsing triple star, which can be ruled out from the radial velocity measurements. ».
Is this the candidate planet GW Ori b ?
(https://ui.adsabs.harvard.edu/abs/2021MNRAS.508..392S/abstract )
I have not found in the literature any paper excluding GW Ori b from RV measurements.
For completeness, it would be useful that the author gives the candidate circum-triple planets name and a reference for its rebuttal.
Can the author clarify this point?
--> Response to reviewer: I apologize for the confusing statement. I meant triply-eclipsing triple stars that produce tertiary eclipses and/or occultations. Footnote 2 has been updated for clarity.
# # #
3) From a general point of view, since the paper intends to be a general review, I make some suggestions:
i) Circumbinary planets are a subclass, generally named “P-type”, of planets in binary star systems, planets orbiting only one star in a wide binary being of “S-type”. The author could place circumbinary planets in perspective in this scheme, and remind that the planetary stable orbits must have a semi-major axis < a/3 for S-type planets and > 3a for P-type planets, where a is the the separation between the two stars (for equal mass stars)
ii) The author could add the method of selecting stars to search for transits by Hale and Doyle:
https://ui.adsabs.harvard.edu/abs/1994Ap%26SS.212..335H/abstract
iii) In the spirit of the general review aspect of the paper, the author could add a few paragraphs on habitability of circumbinary planets: e.g. papers by Eggl and by Haghighipour
--> Response to reviewer: The introduction has been expanded to include a brief discussion of the P-, S-, and L-type planets in binary systems. A reference to Hale & Doyle (1994) has been added to the text, as well as a brief discussion on the habitability of circumbinary planet.
# # #
Reviewer 2 Report
The manuscript presents a reasonably compact and complete overview over the status of transiting circumbinary planets, and should be accepted following minor revision, with the comments as follows:
It should be clearly stated – both in abstract (e.g. in 2nd last line) and in the introduction – that this review is (only) about CBPs around main-sequence binaries. Additionally, in the introduction, the potential existence of CBPs around evolved stellar systems found by eclipse timing should at least be mentioned (a good reference is Marsh, 2018haex.bookE..96M).
Further comments coresponding to line-numbers:
48: "make close EBs excellent targets" : 'close EBs' should be specified a bit more precisely; in the context of this work e.g. 'non-contact binaries with periods from days to a few months'
53: where TEP is mentioned the first time, the more relevant papers 8, 70 should also be cited.
91: Give the ref for the 3rd Kepler-47 CBP (Oscoz+2019)
101: "Two non-transiting CBPs [add:] around main-sequence binaries have been discovered as well"
HD 202206 c, discovered by RVs, should also be noted, which is a borderline case for a CBP with a G+M MS-binary and the 3rd object a borderline object between planet and BD.
114: the critical distance a_crit should be defined, and its approximate size given (e.g. a_crit/a_bin ~2.3 from Dvorak+1989, Holman&Wiegert 1999, for coplanar case and e_binary=0). Also, the corresponding period-ratio: P_crit/P_bin ~3.5 should be indicated.
115: "One intriguing consequence": No-one really expected to ever discover USP equivalents for CBPs, because of P_crit ~3.5x P_bin. It would be better to state that the shortest expected CBPs were in the P= 3-4 days range, given that non-contact/non-evolved MS binaries have periods of >~ 1d.
[A method to find such short-period CBPs from reflected light of the binary on the planet was once devised, Deeg & Doyle 2011EPJWC..1105005D, but never applied once it got clear that these planets are unlikely to exist. sFeel free to add this]
Fig. 2: The red crosses (stability limit) should be mentioned in the caption. Also: 'The different colours are to be ignored' (the meaning of grey vs orange is neither obvious in the original paper)
Fig.4 : left panel: the caption is not really comprehensible without looking into the original ref; so please add at least the meaning of the black line that falls below the x-axis.
202: Pls add something like this: The absence of CBPs with P<7.5d was also corroborated by a non-detection of CBPs in the CoRoT mission sample, which had a sample size similar to Kepler, but with shorter coverages, which would however been sensitive to the detection of short-periodic CBPs [107] (Klagyivik+). .
247: ...by visual inspection of EB lightcurves, [pls add something like this:] which is typically done by visual inspection of a lightcurve that had been 'cleaned' from low-frequent flux-variations and that was then folded by the period of the binary.
267- 282: The detailed treatment of these 3 false positives is out of place; these lines should be removed. However, (some of) this text could be integrated into the captions of Fig. 8,9,10.
288: To ref 113 , PLATO, you might add that a several-times (e.g. 3-5 x) increase of the current CBP sample is expected.
295: The K2 -> the K2.
It is also suggested to move this paragraph about K2 to an earlier place, lines 290 -294 make for a better ending.
References:
At the time of this review, none of the links to arxiv preprints worked. A message "System Unavailable. We are currently performing scheduled maintenance." shows up. Arxiv has also stated that mirrors will be discontinued, see https://info.arxiv.org/help/mirrors.html
Hence, all links to xxx.lanl.gov should be replaced by links to arxiv.org
Minor typos and style issues:
Abstract, first line: 'oversized' may have a negative connotation here, as in 'larger than corresponds to them'. 'far-reaching, paramount or supreme seem better
27: signatures with no -> signature without
32 star then migrate -> star and then are thought to migrate [its a theory]
46 Elsender et al -> convert to numbered citation
55 and several times more: MA, MB, MSun, R1, R2, RSun : These should be subscripted, M_A, M_{Sun} etc. Also use R_A and R_B when the mass uses an index A and B.
97: With these -> with all of these discoveries [else it seems to refer to K-413 of the previous phrase]
164: While the 12 known CBP....
168: proximity to the stability....
188: Compared to -> Within the
Fig. 9: 2nd line: ...of the TESS data, showing ... (remove 'for the')
The manuscript has a number of minor failures; but is of sufficient quality for a scientific journal.
Author Response
Oct 15, 2023
Dear reviewer,
Thank you for the detailed feedback. I have revised the manuscript "Transiting Circumbinary Planets in the Era of Space-Based Photometric Surveys", following closely your suggestions.
Please find below the detailed response to your comments. In the revised manuscript, I have set off updated or new text in bold faced font for your convenience.
Sincerely,
Veselin Kostov
--------------
The manuscript presents a reasonably compact and complete overview over the status of transiting circumbinary planets, and should be accepted following minor revision, with the comments as follows:
# # #
It should be clearly stated – both in abstract (e.g. in 2nd last line) and in the introduction – that this review is (only) about CBPs around main-sequence binaries.
--> Response to reviewer: Clarifying statements added to the abstract and the introduction.
# # #
Additionally, in the introduction, the potential existence of CBPs around evolved stellar systems found by eclipse timing should at least be mentioned (a good reference is Marsh, 2018haex.bookE..96M).
--> Response to reviewer: Done
# # #
Further comments coresponding to line-numbers:
48: "make close EBs excellent targets" : 'close EBs' should be specified a bit more precisely; in the context of this work e.g. 'non-contact binaries with periods from days to a few months'
--> Response to reviewer: Text updated accordingly to specify the EB targets.
# # #
53: where TEP is mentioned the first time, the more relevant papers 8, 70 should also be cited.
--> Response to reviewer: Deeg et al. (1998) references added to the first mention of TEP
# # #
91: Give the ref for the 3rd Kepler-47 CBP (Oscoz+2019)
--> Response to reviewer: Reference to Orosz et al. (2019) added
# # #
101: "Two non-transiting CBPs [add:] around main-sequence binaries have been discovered as well"
HD 202206 c, discovered by RVs, should also be noted, which is a borderline case for a CBP with a G+M MS-binary and the 3rd object a borderline object between planet and BD.
--> Response to reviewer: A brief discussion and reference for HD 202206 added to the text.
# # #
114: the critical distance a_crit should be defined, and its approximate size given (e.g. a_crit/a_bin ~2.3 from Dvorak+1989, Holman&Wiegert 1999, for coplanar case and e_binary=0). Also, the corresponding period-ratio: P_crit/P_bin ~3.5 should be indicated.
--> Response to reviewer: The text has been updated to define a_crit and provide additional details on the stability limit
# # #
115: "One intriguing consequence": No-one really expected to ever discover USP equivalents for CBPs, because of P_crit ~3.5x P_bin. It would be better to state that the shortest expected CBPs were in the P= 3-4 days range, given that non-contact/non-evolved MS binaries have periods of >~ 1d. 
[A method to find such short-period CBPs from reflected light of the binary on the planet was once devised, Deeg & Doyle 2011EPJWC..1105005D, but never applied once it got clear that these planets are unlikely to exist. sFeel free to add this]
--> Response to reviewer: The entire sentence starting with "One intriguing consequence" was removed and a brief discussion on the detection of CBPs light echos has been added.
# # #
Fig. 2: The red crosses (stability limit) should be mentioned in the caption. Also: 'The different colours are to be ignored' (the meaning of grey vs orange is neither obvious in the original paper)
--> Response to reviewer: Caption of Figure 2 update accordingly.
# # #
Fig.4 : left panel: the caption is not really comprehensible without looking into the original ref; so please add at least the meaning of the black line that falls below the x-axis.
--> Response to reviewer: Caption of Figure 4 updated.
# # #
202: Pls add something like this: The absence of CBPs with P<7.5d was also corroborated by a non-detection of CBPs in the CoRoT mission sample, which had a sample size similar to Kepler, but with shorter coverages, which would however been sensitive to the detection of short-periodic CBPs [107] (Klagyivik+). .
--> Response to reviewer: Done
# # #
247: ...by visual inspection of EB lightcurves, [pls add something like this:] which is typically done by visual inspection of a lightcurve that had been 'cleaned' from low-frequent flux-variations and that was then folded by the period of the binary.
--> Response to reviewer: Done
# # #
267- 282: The detailed treatment of these 3 false positives is out of place; these lines should be removed. However, (some of) this text could be integrated into the captions of Fig. 8,9,10.
--> Response to reviewer: The lines have been removed and some of the text has been integrated into the figure captions.
# # #
288: To ref 113 , PLATO, you might add that a several-times (e.g. 3-5 x) increase of the current CBP sample is expected.
--> Response to reviewer: Done. However, the only reference I could find is from "Circumbinary planet (CBP) detection: implications for the PIC", at https://indico.ict.inaf.it/event/806/contributions/4313/attachments/2657/5210/20190925_Deeg.pdf.
# # #
295: The K2 -> the K2. 
It is also suggested to move this paragraph about K2 to an earlier place, lines 290 -294 make for a better ending.
--> Response to reviewer: the text has been rearranged such that the K2 paragraph swapped places with the current last sentence.
# # #
References:
At the time of this review, none of the links to arxiv preprints worked. A message "System Unavailable. We are currently performing scheduled maintenance." shows up. Arxiv has also stated that mirrors will be discontinued, see https://info.arxiv.org/help/mirrors.html
Hence, all links to xxx.lanl.gov should be replaced by links to arxiv.org
--> Response to reviewer: Fixed.
# # #
Minor typos and style issues:
Abstract, first line: 'oversized' may have a negative connotation here, as in 'larger than corresponds to them'. 'far-reaching, paramount or supreme seem better
--> Response to reviewer: 'oversized' replaced with 'paramount'
# # #
27: signatures with no -> signature without
--> Response to reviewer: Done
# # #
32 star then migrate -> star and then are thought to migrate [its a theory]
--> Response to reviewer: Done
# # #
46 Elsender et al -> convert to numbered citation
--> Response to reviewer: Fixed
# # #
55 and several times more: MA, MB, MSun, R1, R2, RSun : These should be subscripted, M_A, M_{Sun} etc. Also use R_A and R_B when the mass uses an index A and B.
--> Response to reviewer: Done
# # #
97: With these -> with all of these discoveries [else it seems to refer to K-413 of the previous phrase]
--> Response to reviewer: Done
# # #
164: While the 12 known CBP....
--> Response to reviewer: Done
# # #
168: proximity to the stability....
--> Response to reviewer: Done
# # #
188: Compared to -> Within the
--> Response to reviewer: Done
# # #
Fig. 9: 2nd line: ...of the TESS data, showing ... (remove 'for the')
--> Response to reviewer: Done
# # #
Reviewer 3 Report
universe 2670507 Transiting Circumbinary Planets in the Era of Space-Based Photometric Surveys
last paragraph of Section 1:
you state "A closer inspection of all currently-available TESS data shows no obvious transits"
To clarify you should add on line 64 that primary and secondary eclipses are for the two stars.
line 69: Why is the linear ephemeris interesting? If not, omit the statement.
Fig.2 caption: "The horizontal bars in the left panel
highlight, on a logarithmic scale, the orbital eccentricity of the planets and of their binary hosts." should be changed to:
"The horizontal bars in the right panel show, on a logarithmic scale, the orbital eccentricity of the planets (bars at large distance) and of their binary hosts (bars at smaller distance)."
Also explain the color scheme.
line 112: KIC 7821010 is equally far away, so should be included.
line 121: "right panel" not left
line 188: "CBPs are fairly common" is an ambiguous description: do you mean they "are normal stars" or "normal binaries" or other? (fairly common usually means in large number or percent of the population)
line 286: avoid the term "the great observatories" (replace the word great) because that has special meaning in astronomy.
minor comments:
line 27: "signature"
line 31: error in reference numbering (15 is listed twice).
line 40: "has been" should be "was"
line 46: reference should be numbered
line 55: change RSun, MSun to proper notation $R_{\odot}$, $M_{\odot}$
Line 61: change "Speaking of TESS, the telescope" to "TESS"
line 62: "of which are in"
line 108 "as discussed below"
line 198 "With that said" adds nothing and should be omitted.
line 200: "periods"
line 236: reference should be numbered
line 259, line 298 and line 298: either replace the "etc" an particular explanation or omit it.
The comments on corrections to English language are in the minor comments section of the review report
Author Response
Oct 15, 2023
Dear reviewer,
Thank you for the detailed feedback. I have revised the manuscript "Transiting Circumbinary Planets in the Era of Space-Based Photometric Surveys", following closely your suggestions.
Please find below the detailed response to your comments. In the revised manuscript, I have set off updated or new text in bold faced font for your convenience.
Sincerely,
Veselin Kostov
--------------
last paragraph of Section 1:
you state "A closer inspection of all currently-available TESS data shows no obvious transits"
To clarify you should add on line 64 that primary and secondary eclipses are for the two stars.
--> Response to reviewer: Clarifying statement added.
# # #
line 69: Why is the linear ephemeris interesting? If not, omit the statement.
--> Response to reviewer: Statement removed.
# # #
Fig.2 caption: "The horizontal bars in the left panel highlight, on a logarithmic scale, the orbital eccentricity of the planets and of their binary hosts." should be changed to:
"The horizontal bars in the right panel show, on a logarithmic scale, the orbital eccentricity of the planets (bars at large distance) and of their binary hosts (bars at smaller distance)."
Also explain the color scheme.
--> Response to reviewer: Figure caption updated accordingly, color scheme explained
# # #
line 112: KIC 7821010 is equally far away, so should be included.
--> Response to reviewer: KIC 7821010 added to the statement
# # #
line 121: "right panel" not left
--> Response to reviewer: The orbital periods are shown in the left panel of Figure 2. The right panel shows the orbital separations.
# # #
line 188: "CBPs are fairly common" is an ambiguous description: do you mean they "are normal stars" or "normal binaries" or other? (fairly common usually means in large number or percent of the population)
--> Response to reviewer: Statement updated to clarify that the CBP hosts are typical in terms of stellar mass, mass ration, orbital period, etc.
# # #
line 286: avoid the term "the great observatories" (replace the word great) because that has special meaning in astronomy.
--> Response to reviewer: "great" removed from the description
# # #
minor comments:
line 27: "signature"
--> Response to reviewer: Fixed
# # #
line 31: error in reference numbering (15 is listed twice).
--> Response to reviewer: Reference fixed
# # #
line 40: "has been" should be "was"
--> Response to reviewer: Done
# # #
line 46: reference should be numbered
--> Response to reviewer: Fixed
# # #
line 55: change RSun, MSun to proper notation $R_{\odot}$, $M_{\odot}$
--> Response to reviewer: Fixed
# # #
Line 61: change "Speaking of TESS, the telescope" to "TESS"
--> Response to reviewer: Done
# # #
line 62: "of which are in"
--> Response to reviewer: Fixed
# # #
line 108 "as discussed below"
--> Response to reviewer: Fixed
# # #
line 198 "With that said" adds nothing and should be omitted.
--> Response to reviewer: Done
# # #
line 200: "periods"
--> Response to reviewer: Fixed
# # #
line 236: reference should be numbered
--> Response to reviewer: Fixed
# # #
line 259, line 298 and line 298: either replace the "etc" an particular explanation or omit it.
--> Response to reviewer: "etc" removed
# # #